# Susceptibility toward cefiderocol and sulbactam-durlobactam in extensively drug-resistant *Acinetobacter baumannii* detected from ICU admission screening in Hanoi, Vietnam, 2023

Sébastien Boutin,[1,2,3] Nguyen Quang Toan,[4,5] Thi Anh Mai Pham,[1,4] Truong Nhat My,[4,5] Nguyen Thi Kim Phuong,[5] Bui Tien Sy,[4,5] Nguyen Van Trong,[5] Lisa Göpel,[1] Leo Huber,[1] Kaan Kocer,[1,2] Le Thi Kieu Linh,[4,6] Tran Thanh Tung,[1] Nguyen Trong The,[4,5] Le Huu Song,[4,5,6] Thirumalaisamy P. Velavan,[4,6,7] Dennis Nurjadi[1,2,4]

**ABSTRACT** Infections due to carbapenem-resistant *Acinetobacter baumannii* (CRAB) pose significant clinical challenges due to limited treatment options. In this study, we investigated the molecular epidemiology, resistance profiles, and genomic characteristics of CRAB isolates from intensive care unit (ICU) admission screenings in Vietnam, focusing on susceptibility to the novel agents cefiderocol and sulbactam-durlobactam. Between 1 July and 31 October 2023, extended antibiotic susceptibility testing (AST) and whole-genome sequencing were performed on CRAB isolates obtained through ICU admission and weekly screenings at the 108 Military Central Hospital in Hanoi. CRAB colonization was detected in 31 of 691 ICU patients (4.5%), with 30 isolates classified as multidrug-resistant/extensively drug-resistant and an ST[pas]164 isolate resistant to all substances tested except for sulbactam-durlobactam. All isolates were resistant to ciprofloxacin, levofloxacin, imipenem, and meropenem, with high resistance rates to amikacin (96.8%) and trimethoprim/sulfamethoxazole (96.8%). Cefiderocol resistance was found in four ST[pas]164 isolates and one ST[pas]338-1LV (16.1%), displaying alterations in *pirA* and *piuA* genes, while sulbactam-durlobactam non-susceptibility (61.3%) was observed in isolates carrying PBP3 alterations. Molecular characterization revealed high-risk clades, with ST[pas]2 as the most prevalent, followed by the emerging ST[pas]164 and ST[pas]16 clones, which showed significant resistance and virulence potential. CRAB prevalence among ICU patients and the emergence of highly resistant clones show the need for surveillance and alternative therapeutic options. The presence of cefiderocol resistance and the high rate of sulbactam-durlobactam non-susceptibility in CRAB isolates without prior exposure raise concerns about the spread of resistance in asymptomatic carriers. AST should be performed before using novel antibiotic agents.

**IMPORTANCE** Carbapenem-resistant *Acinetobacter baumannii* (CRAB) is a highly drug-resistant bacterium that poses a serious threat in hospitals, especially to patients in intensive care units (ICUs). This study examined CRAB bacteria found in ICU patients in Vietnam, focusing on their resistance to antibiotics, including new antibiotics such as cefiderocol and sulbactam-durlobactam. The results showed that CRAB in Vietnam is resistant to almost all the antibiotics tested, making infections extremely difficult to treat. Alarmingly, some bacteria were resistant to cefiderocol and sulbactam-durlobactam even in patients who had never received these drugs, suggesting that resistance is spreading quietly. This highlights the urgent need for ongoing surveillance, early detection, and careful use of antibiotics to prevent the spread of untreatable infections.

**KEYWORDS** carbapenem-resistant *Acinetobacter baumannii*, Vietnam, admission screening, genomic investigation, cefiderocol

Address correspondence to Dennis Nurjadi, dennis.nurjadi@uni-luebeck.de.

Sébastien Boutin and Nguyen Quang Toan contributed equally to this article. Shared authors are listed in alphabetical order by surname.

D.N. received speakers honoraria from Shionogi and Cepheid and has participated on the advisory board for Shionogi, all outside the scope of this work.

See the funding table on p. 9.

The study was presented as an e-poster at ESCMID Global 2025 (annual conference of the European Society of Clinical Microbiology and Infectious Diseases) in Vienna, Austria.

The rapid rise of antibiotic resistance poses a major challenge in modern medicine. Among the high-priority pathogens identified by the WHO, *Acinetobacter baumannii* stands out with its alarming resistance to carbapenems and antibiotics of last resort (1). *A. baumannii* is a gram-negative coccobacillus that acts as an opportunistic pathogen, causing nosocomial infections such as pneumonia, bacteremia, meningitis, and/or skin or soft tissue infections (2, 3). The increasing prevalence of multidrug-resistant (MDR) *A. baumannii* infections, along with the lack of antimicrobial options, complicates clinical management (4, 5), especially in countries with limited resources. In addition, *A. baumannii* exhibits intrinsic resistance to disinfectants, enabling its long-term survival on surfaces that act as reservoirs, thereby contributing to nosocomial infections in healthcare settings (6).

The global rise in carbapenem resistance in *A. baumannii* has necessitated the development of novel antibiotics, such as cefiderocol and sulbactam-durlobactam (7–10), which offer new mechanisms of action to combat MDR pathogens. Cefiderocol is a siderophore-conjugated cephalosporin with side chain modifications, which can protect this substance from hydrolysis by metallo-beta-lactamases (11). Durlobactam is a potent inhibitor of class D beta-lactamases, including carbapenemases of the OXA family (OXA-23, OXA-24/40), which are commonly encountered in carbapenem-resistant *Acinetobacter baumannii* (CRAB) (8).

Although admission screening data can provide valuable insights into the prevalence and molecular epidemiology of MDR pathogens in the community, information specific to CRAB in Vietnam remains scarce. This study aims to assess the susceptibility of CRAB isolates from admission screenings in Vietnam to cefiderocol and sulbactam-durlobactam, while also investigating the molecular characteristics by whole-genome sequencing. As of November 2024, neither cefiderocol nor sulbactam-durlobactam is available or used in clinical practice in Vietnam, making it interesting to determine whether resistance to these agents already exists in CRAB isolates.

## MATERIALS AND METHODS

### Study design

In this prospective cohort study, we conducted admission screening and weekly screening for multidrug-resistant gram-negative (MDRGN) until discharge for patients admitted to the intensive care units (ICUs) between 1 July and 30 October 2023 at the 108 Military Central Hospital in Hanoi, Vietnam. The inclusion criteria were admission to ICU, informed consent, age over 18 years, and MDRGN screening within 48 hours of admission. Of the 691 patients screened in the study, 31 were screened positive for CRAB, and all 31 isolates underwent genotypic and phenotypic characterization. In this study, only the first isolate per patient was included for genome sequencing and analysis.

### Microbiological procedure

Rectal swabs for carbapenem-resistant gram-negative bacilli were collected using eSwabs (Copan, Italy) for further processing in the microbiology laboratory. Briefly, 10 µL of Amies medium was inoculated onto a selective chromogenic medium for the detection of carbapenem-resistant gram-negative bacilli (CHROMagar mSuper-CARBA). After overnight incubation, the plates were examined for microbial growth, and colonies growing on the selective medium were identified using Matrix-Assisted Laser Desorption/Ionization – Time of Flight mass spectrometry (MALDI-TOF MS, Vitek MS) at the microbiology laboratory in Vietnam. Antibiotic susceptibility testing (AST) was performed using the Micronaut MDR gram-negative broth microdilution panel (Bruker, Germany). The antibiotics included in the test panel were amikacin, meropenem, imipenem, colistin, ciprofloxacin, levofloxacin, and trimethoprim/sulfamethoxazole. Other antibiotics for which there were no clinical breakpoints were not included in the analysis. *Escherichia coli* ATCC25922 was used as a quality control strain. Cefiderocol

AST was performed by the disc diffusion method and confirmed by the reference method broth microdilution with iron-depleted cation-adjusted Mueller-Hinton broth as previously published for isolates falling into the area of technical uncertainty category according to European Committee on Antimicrobial Susceptibility Testing (EUCAST) recommendations. Antibiotic susceptibility was interpreted using EUCAST clinical breakpoints version 14.0. Isolates were considered resistant if either the disk diffusion test gave a diameter below 17 mm or the MIC value was higher than 2 µg/mL. Sulbactam-durlobactam was tested using the broth microdilution method as twofold dilutions of sulbactam (MedchemExpress, USA) in combination with a fixed concentration of 4 µg/mL of durlobactam (MedchemExpress, USA), performed in triplicate, and the median MIC was determined (12). The MIC was defined as the lowest concentration inhibiting visual growth. The control strain *A. baumannii* NCTC 13304 was used in parallel for the testing. The median MIC was 2 µg/mL (range 1–2 µg/mL), and the results were interpreted according to the CLSI breakpoints (susceptible ≤4 µg/mL, resistant ≥16 mg/L). Both cefiderocol and sulbactam-durlobactam were not considered for the classification of multidrug-resistant (MDR)/extensively drug-resistant (XDR) (13). MDR was defined as resistance to at least one agent in three or more antimicrobial categories. XDR was defined as resistance to all antimicrobials except polymyxins and tigecycline. Pan resistance was defined as resistance to all antibiotics tested, except for sulbactam-durlobactam.

## Whole-genome sequencing and bioinformatics analysis

DNA extraction was performed using the Qiagen DNeasy Blood and Tissue Kit (Qiagen GmbH, Hilden, Germany) following the manufacturer's instructions. The DNA was used as input for library preparation using the Illumina DNA Prep Kit (Illumina), and sequencing was done on a Nextseq 2000 Illumina platform (short-read sequencing, 2 × 51 bp). Post-sequencing procedure was performed as follows: raw sequences were controlled for quality and adapter removal using fastp (v0.23.2 with parameters $-q = 30$ and $-l = 45$) (14). One pan-resistant isolate (AB16) was subjected to long-read sequencing as well using Ligation Sequencing Kit V14 (Oxford Nanopore Technology, Oxford, United Kingdom) and sequenced on a MinION Mk1B (R10.4.1) sequencer. Data acquisition and basecalling were performed using Dorado (0.7.0) super-accurate mode (simplex barcoding and model: dna_r10.4.1_e8.2_400bps_sup@v4.3.0).

Clean reads were then used to create *de novo* assembly using SPAdes 3.15.5 (with the option—careful and—only-assembler) (15). Draft genomes were curated by removing contigs with a length < 500 bp and/or coverage < 10× . For the isolate AB16, the curated reads from long-read and short-read sequencing were then used together to create a hybrid assembly using Unicycler (v0.5.0) (16) and polish using Polypolish (v0.5.0) (17). The quality of the final draft was quality-controlled using Quast (v5.0.2) (18). The species identification of each draft genome was done using mash (sub-command screen) by querying each draft genome to a database composed of a representative genome of each species present in the Microbial Genomes resource (https://www.ncbi.nlm.nih.gov/genome/microbes/). Multilocus Sequence Typing (MLST) was defined using the tool mlst v2.23.0 which uses the database PubMLST with the Pasteur scheme (Seemann T, Github https://github.com/tseemann/mlst) (19). K and OC loci identification was performed using Kaptive v3.0.0b5 with the Acinetobacter databases (v2.0.4) (20, 21). ANI was calculated using an all-vs-all analysis of the genomes using ANIclustermap (v1.3.0; https://github.com/moshi4/ANIclustermap).

The complete draft genomes were annotated using bakta (v1.9.4) (22) and processed through available databases using AMRFinderPlus and Abricate (minimum identity: 90% and minimum coverage: 80%) (https://github.com/tseemann/abricate) to identify antimicrobial resistance (NCBI, CARD, ARG-ANNOT, ResFinder, MEGARES databases downloaded on 12/07/2024), virulence genes (VFDB databases downloaded on 12/07/2024), and plasmid types (PlasmidFinder database downloaded on 12/07/2024) to identify the Inc type of the plasmid (23–28). Resistance to colistin mediated by mutation

was checked using Breseq (v0.38.3) using the reference strain ATCC 19606 (CP045110) (29). All genomes were mapped and compared to extract only Single Neucleotide Porlymorphisms (SNPs) found in resistant strains as potentially related to antimicrobial resistance.

Each genome was aligned to the representative reference genome (CP045110) using SKA2 v0.3.8 (Split Kmer Analysis) (30), and the alignment was then used as input for Gubbins (v3.3.1) (31) to reduce the effect of recombinations and export SNP table using snp-dist (v0.8.2, https://github.com/tseemann/snp-dists). In the end, 56,549 polymorphic sites were used for the construction of the phylogenetic tree with Ramxl (v8.2.12, model GTRGAMMA, and 10,000 bootstrap iterations) (32). Clonal clusters were defined using both ANI and SNP threshold (ANI ≥ 99.99 and/or SNP distance ≤ 10) (33). The analysis of the amino-acid sequence of *pirA* and *piuA* was performed using the predicted sequence by bakta. Sequences were aligned using mafft (v7.520) (34), and the phylogeny was inferred using fasttree (v2.1.11, JTT+CAT model with 1,000 resamples) (35).

## Statistical analysis

Descriptive statistics were performed using R 4.3.3.

## RESULTS

Between 1 July and 31 October 2023, 691 patients admitted to the ICUs of the 108 Military Central Hospital in Northern Vietnam were screened for asymptomatic MDRGN colonization. Rectal colonization with CRAB was detected in 4.5% (31 of 691 patients). All isolates were non-duplicates (only one isolate per patient was included) and were characterized by short-read genome sequencing. Overall, males were overrepresented with 74.2% (23/31). The median age was 68 years, and the median length of stay was 13 days. Only eight patients (26%) were positive at admission; the other colonization was detected during the weekly follow-up screening.

### Phenotypic susceptibility profile

All the isolates were resistant to ciprofloxacin, levofloxacin, imipenem, and meropenem. In addition, almost all isolates were resistant to amikacin (30/31, 96.8%) and/or trimethoprim/sulfamethoxazole (30/31, 96.8%). Resistance to cefiderocol was observed in five isolates (5/31, 16.1%), while two isolates (2/31, 6.5%) were resistant to colistin. For sulbactam-durlobactam, 6 isolates (6/31, 19.4%) were resistant, 13 (13/31, 41.9%) showed intermediate susceptibility, and 12 (12/31, 38.7%) were susceptible. One isolate (AB16) was resistant to all the substances tested except for sulbactam-durlobactam, leading to an overall detection of 30 MDR/XDR and an isolate belonging to ST$^{pas}$164, harbouring $bla_{OXA-23}$ and $bla_{OXA-91}$, which was resistant to all antibiotics tested (Table 1; Fig. 1).

### Molecular epidemiology and genetic clusters

The most prevalent MLST was ST$^{pas}$2 ($n = 21$), with all isolates carrying the O-locus OCL1 with various K-loci (1 KL116, 4 KL2, 6 KL3, 1 KL40, 3 KL52, and 6 KL6). A total of three clusters (C01–C03) were identified within ST$^{pas}$2. Cluster C01 contained five isolates (KL6-OCL1), where only two patients showed temporal overlap. The clusters C02 (KL52-OCL-1) and C03 (KL03-OCL-1) each contained two isolates, yet the temporal overlap of the patients could not be evaluated due to incomplete data on the ward stay. The less prevalent MLSTs were ST$^{pas}$16 ($n = 3$, KL24, 2 OCL2 and 1 OCL7), ST$^{pas}$164 ($n = 4$, KL47-OCL13). Within ST$^{pas}$164, we observed a cluster C04 containing three isolates but only two patients with temporal overlap. Finally, three more MLSTs were observed with a single occurrence: ST$^{pas}$187 (KL3-OCL1), ST$^{pas}$338-1LV (KL229-OCL14) & ST$^{pas}$571-1LV (KL10-OCL1) (Fig. 1; Fig. S1).

All CRAB carried various beta-lactamase genes, but the carbapenem resistance is due to the presence of either $bla_{OXA-23}$ (oxacillinases) (28/31) or $bla_{NDM-1}$ (New-Delhi-metallo-beta-lactamase) (3/31). The gene $bla_{TEM-1}$ is almost the only variant within the class A beta-lactamases (19/31), and one isolate carried the gene $bla_{PER-1}$ (*Pseudomonas*

**TABLE 1** Antimicrobial susceptibility testing of the *A. baumanii* isolates[a]

| Isolate | Amikacin MIC | Cefiderocol MIC | Cefiderocol DD | Imipenem MIC | Meropenem MIC | Sulbactam MIC | Sulbactam-durolobactam median MIC (range) | Ciprofloxacin MIC | Levofloxacin MIC | Trimethoprim-sulfamethoxazol MIC | Colistin MIC |
|---|---|---|---|---|---|---|---|---|---|---|---|
| AB-01 | >32 | 0.5 | 23 | >8 | 64 | 32 | 4 (4–8) | >2 | >2 | >4/76 | 1 |
| AB-02 | >32 | 0.125 | 24 | >8 | 128 | 64 | 8 (8–16) | >2 | >2 | >4/76 | 1 |
| AB-03 | >32 | 0.25 | 23 | >8 | 64 | 128 | 8 (8–8) | >2 | >2 | >4/76 | 1 |
| AB-04 | >32 | 0.25 | 23 | >8 | 128 | 64 | 4 (4–8) | >2 | >2 | >4/76 | 1 |
| AB-05 | >32 | 1 | 18 | >8 | >128 | 64 | 32 (32–32) | >2 | >2 | >4/76 | 1 |
| AB-06 | >32 | 0.25 | 24 | >8 | 128 | 64 | 8 (8–8) | >2 | >2 | >4/76 | 1 |
| AB-07 | >32 | 0.5 | 21 | >8 | 64 | 64 | 4 (4–4) | >2 | >2 | >4/76 | 1 |
| AB-08 | >32 | 0.5 | 22 | >8 | 64 | 64 | 16 (16–16) | >2 | >2 | >4/76 | 1 |
| AB-09 | >32 | 0.5 | 21 | >8 | 64 | 128 | 16 (16–16) | >2 | >2 | >4/76 | 1 |
| AB-10 | 8 | **32** | 17 | >8 | 32 | 16 | 2 (1–2) | >2 | >2 | 4/76 | 1 |
| AB-11 | >32 | 0.5 | 20 | >8 | 128 | 64 | 8 (8–8) | >2 | >2 | 1/19 | 1 |
| AB-12 | >32 | 0.5 | 19 | >8 | 64 | 64 | 8 (4–8) | >2 | >2 | >4/76 | 1 |
| AB-13 | >32 | 1 | 18 | >8 | 128 | 64 | 32 (32–32) | >2 | >2 | >4/76 | 1 |
| AB-14 | >32 | **8** | **11** | >8 | 32 | 32 | 2 (2–2) | >2 | >2 | 4/76 | 1 |
| AB-15 | >32 | **8** | **12** | >8 | 32 | 16 | 2 (2–4) | >2 | >2 | >4/76 | 1 |
| AB-16 | >32 | **8** | **10** | >8 | >128 | 32 | 1 (0.25–1) | >2 | >2 | >4/76 | >8 |
| AB-17 | >32 | 0.125 | 23 | >8 | >128 | 64 | 8 (8–8) | >2 | >2 | >4/76 | 1 |
| AB-18 | >32 | 0.125 | 27 | >8 | >128 | 64 | 4 (4–4) | >2 | >2 | >4/76 | 2 |
| AB-19 | >32 | 0.25 | 23 | >8 | 64 | 64 | 4 (4–8) | >2 | >2 | 4/76 | 1 |
| AB-20 | >32 | 1 | 18 | >8 | >128 | 32 | 32 (32–32) | >2 | >2 | >4/76 | 1 |
| AB-21 | >32 | 0.125 | 23 | >8 | >128 | 32 | 8 (4–8) | >2 | >2 | >4/76 | 1 |
| AB-22 | >32 | 0.25 | 26 | >8 | >128 | 128 | 8 (8–8) | >2 | >2 | >4/76 | >8 |
| AB-23 | >32 | **4** | **12** | >8 | 64 | 16 | 2 (2–2) | >2 | >2 | >4/76 | 1 |
| AB-24 | >32 | 0.125 | 22 | >8 | 128 | 64 | 4 (4–4) | >2 | >2 | >4/76 | 1 |
| AB-25 | >32 | 0.125 | 23 | >8 | 128 | 32 | 8 (4–8) | >2 | >2 | >4/76 | 1 |
| AB-26 | >32 | 1 | 21 | >8 | 128 | ≥256 | 16 (16–32) | >2 | >2 | >4/76 | 1 |
| AB-27 | >32 | 0.25 | 25 | >8 | 128 | 64 | 8 (8–8) | >2 | >2 | >4/76 | 1 |
| AB-28 | >32 | 0.25 | 22 | >8 | 128 | 128 | 8 (4–8) | >2 | >2 | 4/76 | 1 |
| AB-29 | >32 | 0.25 | 21 | >8 | 128 | 64 | 8 (8–8) | >2 | >2 | 4/76 | 1 |
| AB-30 | >32 | 0.25 | 24 | >8 | 128 | 32 | 8 (8–8) | >2 | >2 | >4/76 | 1 |
| AB-31 | >32 | 0.25 | 22 | >8 | 64 | 64 | 4 (2–4) | >2 | >2 | >4/76 | 1 |

[a]MIC values in bold font are considered as resistant according to the EUCAST clinical breakpoints, except for sulbactam–durlobactam, which was interpreted according to the CLSI breakpoints.

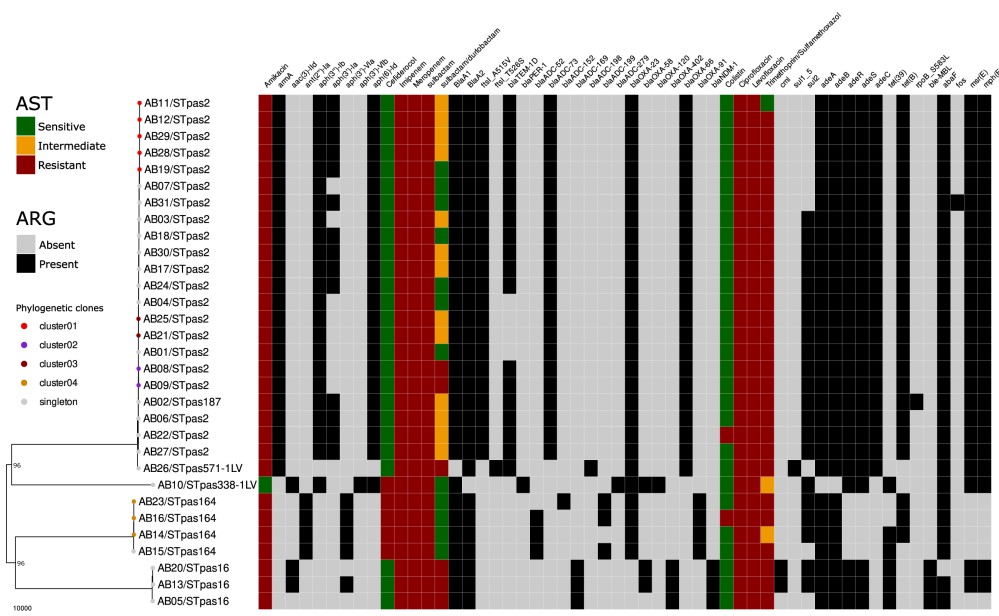

**FIG 1** Phylogenetic analysis and antimicrobial resistance genotype-phenotype of the *A. baumannii* isolates. Phylogenetic tree was obtained using the recombination-corrected whole-genome alignment using Gubbins. Isolates belonging to the same clonal group (cluster based on single nucleotide variants) are coded with identical color-coded dots on the tip of the tree. The phenotypic resistance is color-coded based on EUCAST clinical breakpoint v14.0, and the presence of corresponding antimicrobial resistance genes is symbolized with a black square. ARG, antibiotic resistance genes. Scale of the tree is represented in substitutions per site.

extended resistant). All the CRAB isolates harbored at least one variant of the class C $bla_{ADC}$ with $bla_{ADC-73}$ being the most prevalent (22/31), and one or more class D $bla_{OXA}$ genes ($bla_{OXA-23}$, $bla_{OXA-58}$, $bla_{OXA-51-like}$). The most prevalent $bla_{OXA}$ gene was $bla_{OXA-23}$ (28/31) followed by $bla_{OXA-66}$ (23/31). Interestingly, four of the isolates resistant to cefiderocol harbored the variant $bla_{OXA-91}$. One isolate presented three variants (AB10: $bla_{OXA-23}$,$bla_{OXA-58}$, $bla_{OXA-120}$) while the majority of the remaining isolates exhibited paired carriage of $bla_{OXA-23}$ and $bla_{OXA-51-like}$ (27/31) or $bla_{OXA-58}$ and $bla_{OXA-51-like}$ (2/31). Overall, 23 isolates also present mutations in the gene *fstI*, encoding the transpeptidase region of the penicillin-binding protein 3 (PBP3), 22 A515V and 1 T526S. The isolate with the T526 mutation (AB26) was resistant to sulbactam-durlobactam. Among the isolates with the A515 variant, only two isolates (2/22, 9.1%) were resistant, six isolates (7/22, 31.8%) were susceptible, and 13 (13/22, 59.1%) showed intermediate susceptibility. For this A515V subgroup, the $MIC_{50}$ and $MIC_{90}$ were both 8 µg/mL. The only metallo-beta-lactamase observed in our study population was $bla_{NDM-1}$ (3/31), which was exclusively observed in the $ST^{pas}16$ clade (Fig. 1). As expected, all three $bla_{NDM-1}$-carrying isolates were resistant to sulbactam-durlobactam (10, 36). All five isolates without PBP3 mutation or NDM were susceptible.

Two isolates showed phenotypic resistance to colistin (AB16 and AB22) from phylogenetically different lineages ($ST^{pas}2$ and $ST^{pas}164$). While we observed mutations in the genes previously described in the literature such as *pmrABC* and *LpxCD*, these mutations were not found in the resistant isolates. We found the *pmrB* mutation A138T (also present in 21 other sensitive strains), *pmrC* V42I (also present in 23 other sensitive strains), *pmrC* L150F (present in 22 other sensitive strains), and lpxD E117K (also present in 22 other sensitive strains in the isolate AB22). We also observed the mutation lpxC N287D in both isolates, but it was also present in all isolates from our cohort. A comparison of resistant and sensitive isolates within our cohort highlighted 35 SNPs with non-synonymous effects in 20 genes. Only two genes were affected in both isolates (*bap* and FQU82_026659), but none of the mutations affected known genes involved

in resistance to colistin. Analysis to detect new variants of *mcr* genes by lowering the identity and coverage threshold did not lead to the identification of *mcr*-like genes in those isolates.

The appearance of cefiderocol resistance is either due to the presence of the gene $bla_{PER-1}$ or phylogenetically related to the $ST^{pas}164$ in our cohort, where it may be mediated by the combined alteration of the gene *piuA* via deletion of 10 nt at position 1343 leading to a premature stop codon at aa position 448 and the gene *pirA* via the insertion of the transposon ISAba1 leading to a premature stop codon at position 194 (Fig. S2). As the gene sequence of *pirA* is lineage-specific, we could hypothesize that the sequence of the gene within the phylogenetic branch $ST^{pas}164$ is more likely to be affected by the ISAba1 transposon. The $ST^{pas}16$ isolates showed reduced susceptibility toward cefiderocol, which may be associated with the presence of the $bla_{NDM-1}$ gene (disk diffusion: 18 mm). Furthermore, all isolates were acquired during the hospital stay, suggesting a nosocomial origin and transmission.

## DISCUSSION

The incidence of nosocomial CRAB infections is increasing rapidly in Southeast Asia with prevalence in ICU patients with ventilator-associated pneumonia or bloodstream infection reaching up to 91% (37). A previous study from Vietnam reported a CRAB prevalence of 70.1%, and only about 50% of these infections were acquired during hospitalization. This concerning trend highlights the need to investigate asymptomatic carriers as a potentially overlooked reservoir for CRAB in this region. Further, CRAB colonization may be an important risk factor for subsequent infections and may contribute to the high incidence of nosocomial CRAB infections in this setting (2, 6, 38).

Although only 4.5% of patients admitted to the ICU in Vietnam were found to be positive for CRAB during admission screening, nearly all isolates (30 out of 31) could be classified as at least MDR. The main mechanism of carbapenem resistance observed was the carriage of $bla_{OXA-23}$. All isolates were MDR, confirming the surge in extensive drug resistance in Vietnam (39). This rise in MDR is likely linked to the spread of the $ST^{pas}2$ lineage, a high-risk clone that has spread widely in Vietnam and worldwide (37, 40–43). Furthermore, we identified a pan-resistant isolate belonging to $ST^{pas}164$, which exhibited resistance to all clinically available antimicrobial agents tested. However, this isolate remained susceptible to sulbactam-durlobactam.

Cefiderocol and sulbactam-durlobactam are promising treatment options for CRAB infections, and in our study, we evaluated the prevalence of resistance to both substances among our isolates. Five isolates were phenotypically resistant to cefiderocol, and interestingly, four belonged to the same clade ($ST^{pas}164$). The genomic analysis suggested that the resistance was very likely to be mediated by the alteration of both *pirA* and *piuA* genes, as all the resistant strains presented a truncation of the *piuA* due to a 10 nt deletion and a truncation of the PirA protein due to the transposition of ISAba1 in the *pirA* gene, which may lead to less efficient uptake of cefiderocol by the siderophore receptors. This hypothesis is concordant with other observations involving those two genes in the resistance to cefiderocol in *A. baumannii* (44, 45). As those isolates were all belonging to the $ST^{pas}164$ and all carrying the gene $bla_{OXA-91}$, we cannot rule out lineage dependency or the impact of the beta-lactamase genes. The resistance on the fifth isolate was mediated by the co-carriage of $bla_{PER-1}$ and $bla_{OXA-23}$ as previously described (46).

The proportion of non-susceptibility to sulbactam-durlobactam in our cohort (19/31, 61.3%) was significantly higher compared to previous studies (36), likely due to the high prevalence of PBP3 mutations in our cohort. The role of the A515V PBP3 mutation in sulbactam-durlobactam remains inconclusive. While several studies have reported this mutation in isolates with elevated sulbactam-durlobactam MICs (47, 48), isogenic background studies suggest little to no direct effect on resistance (49). Two of our resistant isolates and all 13 intermediate-susceptible isolates carried this mutation;

notably, all 15 isolates belong to the same clone, which represents a limitation of our study, as clonal effects cannot be ruled out. Further research is needed to clarify the role of A515V in sulbactam-durlobactam resistance. In contrast, the role of the T526S mutation, found in one resistant isolate, in sulbactam-durlobactam resistance has been confirmed in multiple studies, including those using isogenic backgrounds (49). The three isolates with the highest MIC for sulbactam-durlobactam all harbored $bla_{NDM-1}$, which was expected since NDM-type carbapenemases cannot be inhibited by durlobactam, as published by the company that developed durlobactam (50). Further-more, several studies have reported that NDM-positive A. baumannii is resistant to sulbactam-durlobactam (10, 36). Observing such a high non-susceptibility rate in a cohort without prior exposure to this novel agent is particularly concerning and warrants further surveillance.

Our study suggests that XDR ST$^{pas}$164 A. baumannii is emerging in Vietnam. While ST$^{pas}$2 remains one of the most prevalent and globally circulating clones (40–43), the growing reports of drug-resistant ST$^{pas}$164 A. baumannii across multiple countries raise significant concerns. ST$^{pas}$164 is a widely disseminated clone identified in various studies as a risk due to its spread and MDR phenotype (51, 52), particularly prevalent in Asia and Africa (53, 54). Therefore, it is essential to closely monitor the further expansion of this clone. Additionally, our findings show a notable prevalence of ST$^{pas}$16, which aligns with other reports from Southeast Asia that link it to an XDR phenotype (37, 55).

Given the high prevalence of MDR Gram negatives in Vietnam, antibiotic therapy options are severely limited, resulting in a reliance on colistin for treating MDR gram-negative infections. Therefore, due to the high usage of colistin especially in animal husbandry (56), we anticipated a high prevalence of colistin resistance as observed in another study from the same region focusing on K. pneumoniae (57). However, we observed only two occurrences of colistin resistance: one isolate in the ST$^{pas}$2 lineage and another from ST$^{pas}$164. Unfortunately, we could not determine the exact underlying mechanism for colistin resistance. The genomic analysis did not reveal any mutations exclusive to the resistant isolates, suggesting that the resistance could be due to a regulatory process affecting the transport or modification of LPS.

Our study has certain limitations, including its monocentric design and the lack of comprehensive clinical data for some patients. Nevertheless, we identified several putative transmission clusters, indicating that nosocomial transmission of CRAB may occur undetected, resulting in asymptomatic carriers. Given that a significant proportion of colonized individuals (11 out of 28, 39%) subsequently developed CRAB infections, the potential role of asymptomatic carriers warrants further investigation in larger cohorts. Despite the small sample size, our findings provide valuable insights into the molecular epidemiology of CRAB and its non-susceptibility to novel antibiotics in a high AMR-bur-den setting like Vietnam.

In conclusion, our study identified a high prevalence of sulbactam-durlobactam non-susceptibility in CRAB isolates from a sulbactam-durlobactam naïve population as the combination is not used in Vietnam. This resistance is likely attributable to wide-spread and extensive exposure to beta-lactam antibiotics, raising concerns about the potential loss of this therapeutic option even before its clinical approval in Vietnam. Cefiderocol remains a viable therapeutic alternative; however, susceptibility testing is crucial before initiating treatment with last-resort antibiotics. The emergence of cefiderocol-resistant ST$^{pas}$164 CRAB calls for close monitoring to determine if cefiderocol resistance is a characteristic feature of this clone. Vigilance in tracking and mitigating the spread of this clone in the community is essential to preserve the efficacy of novel antibiotics.

## ACKNOWLEDGMENTS

We thank Melanie Albrecht for her assistance in performing WGS. We thank Dr. Paul Higgins for kindly providing us with the A. baumannii NCTC 13304.

The study was funded through grants from the PAN-ASEAN Coalition for Epidemic and Outbreak Preparedness (PACE-UP; DAAD Project ID: 57592343).

Study conception: D.N., S.B., T.P.V.; sample collection, microbiology, and data collection in Vietnam: T.A.M.P., T.N.M., B.T.S., L.T.K.L., N.T.T., L.H.S.; microbiology and confirmation of AST in Germany: L.G., L.H., D.N.; sequencing: S.B., T.T.T.; bioinformatics analysis: S.B.; statistics: S.B.; first draft: S.B., D.N., T.P.V; finalization of manuscript: all authors.

## AUTHOR AFFILIATIONS

[1]Institute of Medical Microbiology and Center for Infectious Diseases, University of Lübeck and University Hospital Schleswig-Holstein Campus Lübeck, Lübeck, Germany
[2]German Center for Infection Research (DZIF), Partner Site Hamburg-Lübeck-Borstel-Riems, Lübeck, Germany
[3]Airway Research Center North (ARCN), Member of the German Center for Lung Research (DZL), Lübeck, Germany
[4]Vietnamese-German Centre for Medical Research (VG-CARE), Hanoi, Vietnam
[5]108 Military Central Hospital, Hanoi, Vietnam
[6]Institute of Tropical Medicine, University of Tübingen, Tübingen, Germany
[7]Faculty of Medicine, Duy Tan University, Da Nang, Vietnam

## AUTHOR ORCIDs

Truong Nhat My  http://orcid.org/0000-0003-2436-7897
Kaan Kocer  http://orcid.org/0000-0002-1134-4197
Thirumalaisamy P. Velavan  http://orcid.org/0000-0002-9809-9883
Dennis Nurjadi  http://orcid.org/0000-0002-1278-5939

## FUNDING

| Funder | Grant(s) | Author(s) |
| --- | --- | --- |
| Deutscher Akademischer Austauschdienst | 57592343 | Thirumalaisamy P. Velavan |

## DATA AVAILABILITY

The draft genomes presented in this study can be retrieved from the NCBI Genbank repositories under the Bioproject PRJNA1195824. The accession number(s) can be found in the Dataset S1.

## ETHICS APPROVAL

The study was approved by the Institutional Review Board of the 108 Military Central Hospital, Hanoi, Vietnam (108MCH/RES/AMR-HAI-V-D2-05-07-2023).

## ADDITIONAL FILES

The following material is available online.

### Supplemental Material

**Dataset S1 (Spectrum00832-25-s0001.xlsx).** Sequencing statistics, meta data, and antimicrobial susceptibility.
**Supplemental material (Spectrum00832-25-s0002.docx).** Table S1, and Fig. S1 and S2.

### Open Peer Review

**PEER REVIEW HISTORY (review-history.pdf).** An accounting of the reviewer comments and feedback.

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
