## [Reviewer comments · Microbiology Spectrum]

Microbiology Spectrum

Susceptibility towards cefiderocol and sulbactam-durlobactam in extensively drug-resistant *Acinetobacter baumannii* detected from ICU admission screening in Hanoi, Vietnam, 2023

Sébastien Boutin, Nguyen Quang Toan, Thi Anh Mai Pham, Truong Nhat My, Nguyen Thi Kim Phuong, Bui Tien Sy, Nguyen Van Trong, Lisa Göpel, Leo Huber, Kaan Kocer, Le Thi Kieu Linh, Thanh Tung Tran, Nguyen Trong The, Le Huu Song, Thirumalaisamy Velavan, and Dennis Nurjadi

Corresponding Author(s): Dennis Nurjadi, Universitat zu Lubeck

Review Timeline:

Submission Date:	March 22, 2025
Editorial Decision:	March 27, 2025
Revision Received:	March 27, 2025
Accepted:	April 30, 2025

Editor: Cezar Khursigara

Reviewer(s): The reviewers have opted to remain anonymous.

Transaction Report:

DOI: <https://doi.org/10.1128/spectrum.00832-25>

Re: Spectrum00832-25 (**Susceptibility towards cefiderocol and sulbactam-durlobactam in extensively drug-resistant *Acinetobacter baumannii* detected from ICU admission screening in Hanoi, Vietnam, 2023**)

Dear Prof. Dennis Nurjadi:

Thank you for the privilege of reviewing your work. Below you will find my comments, instructions from the Spectrum editorial office, and the reviewer comments.

I am pleased to inform you that your manuscript has been editorially accepted for publication. However, there are a few additional questions in the submission form that need to be answered before the final decision. Once these are completed, please return your submission so that I can move your paper forward to acceptance.

Sincerely,
Cezar Khursigara
Editor
Microbiology Spectrum

Re: Spectrum00832-25R1 (**Susceptibility towards cefiderocol and sulbactam-durlobactam in extensively drug-resistant *Acinetobacter baumannii* detected from ICU admission screening in Hanoi, Vietnam, 2023**)

Dear Prof. Dennis Nurjadi:

Your manuscript has been accepted, and I am forwarding it to the ASM production staff for publication. Your paper will first be checked to make sure all elements meet the technical requirements. ASM staff will contact you if anything needs to be revised before copyediting and production can begin. Otherwise, you will be notified when your proofs are ready to be viewed.

Sincerely,
Cezar Khursigara
Editor
Microbiology Spectrum